# Unravelling the skills of data scientists: A text mining analysis of Dutch university master programs in data science and artificial intelligence

**Mathijs J. Mol**[1], **Barbara Belfi**[2], **Zsuzsa Bakk**[3]*

**1** Gastroenterology and Hepatology (AMC), Amsterdam University Medical Center, Amsterdam, North Holland, The Netherlands, **2** ROA, Maastricht University, Maastricht, Limburg, The Netherlands, **3** The unit of Methodology and Statistics, Institute of Psychology, Leiden University, Leiden, South Holland, The Netherlands

* z.bakk@fsw.leidenuniv.nl

**Data Availability Statement:** All the data and code for the analysis is openly available on OSF: https://doi.org/10.17605/OSF.IO/96MQ3.

## Abstract

The growing demand for data scientists in both the global and Dutch labour markets has led to an increase in data science and artificial intelligence (AI) master programs offered by universities. However, there is still a lack of clarity regarding the specific skills of data scientists. This study addresses this issue by employing Correlated Topic Modeling (CTM) to analyse the content of 41 master programs offered by 11 Dutch universities and an interuniversity combined program. We assess the differences and similarities in the core skills taught by these programs, determine the subject-specific and general nature of the skills, and provide a comparison between the different types of universities offering these programs. Our analysis reveals that data processing, statistics, research, and ethics are the core competencies in Dutch data science and AI master programs. General universities tend to focus on research skills, while technical universities lean more towards IT and electronics skills. Broad-focussed data science and AI programs generally concentrate on data processing, information technology, electronics, and research, while subject-specific programs give priority to statistics and ethics. This research enhances the understanding of the diverse skills of Dutch data science graduates, providing valuable insights for employers, academic institutions, and prospective students.

## 1. Introduction

Data science is an interdisciplinary field that leverages computational methods and techniques to derive actionable insights from large, complex datasets [1]. The rise of big data and the growing use of artificial intelligence (AI) have created an enormous need for professionals with advanced data science abilities [2]. These market changes are strongly driven by theoretical developments, namely the development of large language models [3] and generative adversarial networks [4]. These new modeling developments have also led to a need to adjust

**Funding:** The author(s) received no specific funding for this work.

**Competing interests:** The authors have declared that no competing interests exist.

traditional data driven master programs to encompass these new theoretical developments, and to integrate theoretical knowledge into interdisciplinary master programs. The rapid development of these new modeling approaches represents a continuous challenge to data science master programs to adopt their curricula.

Despite the increasing significance of data scientists in today's society, there is still a noticeable lack of clarity regarding the exact skills and expertise these professionals posses [5, 6]. One reason for this is that the master programs that train students to become data scientists differ greatly in curriculum and focus [5, 7, 8]. For example, the skills gained from a biomedical data science program may vastly differ from those acquired in a marketing analytics program. At the same time, [9] state that the general demand of data scientists on the job market has increased by 663% from 2013 to 2018 and the demand of marketing data analysts has increased with 194% in the same timeframe.

Understanding the skills that data scientists acquire during their education is essential for multiple reasons. First, universities and other educational institutions can utilize this information to refine and update their curricula, ensuring they remain aligned with current industry needs and expectations. Second, businesses and organizations seeking to recruit data scientists can better assess the qualifications of candidates based on their understanding of the typical skills possessed by graduates of these programs. Finally, current and aspiring data scientists can use these findings to identify any gaps in their own skills and guide their professional development efforts.

As stated before, to the best of our knowledge, as of now, there is no universally accepted, centralized definition of skills or knowledge maps for the field of data science or AI. A few recent articles have attempted to define the skills required in data science and AI, but these efforts remain scattered and inconsistent. One such example is the Initiative for Analytics and Data Science Standards (IADSS), which has laid the groundwork for a knowledge framework for the discipline [10]. The IADSS framework theoretically identifies key areas of expertise, such as data engineering, data analysis, data visualization, and machine learning, and their corresponding skills. This initiative aims to standardize the understanding of skills required for data science and AI professionals and provides a foundation for future curriculum development and workforce training. In contrast, other authors have taken a more data-driven approach to defining the skills needed in data science and AI, by analysing job descriptions and identifying patterns and trends [11, 12]. This approach offers valuable insights into the real-world demands and expectations of employers in the field, making it more closely aligned with current industry needs. These studies have highlighted the importance of not only technical skills, such as programming languages, machine learning frameworks, and big data tools, but also soft skills, such as communication, teamwork, and problem-solving abilities. Despite these efforts, the lack of a unified and comprehensive definition of skills in data science and AI remains a challenge.

The primary objective of the present study is to gain more clarity in the skills of Dutch data scientists by analysing the content of Dutch university master programs in data science and AI. We chose to focus on these study programs due to their explicit commitment to providing students with advanced data science skills. Our investigation employs text-mining techniques, specifically Correlated Topic Modeling (CTM; [13]), to systematically evaluate the course offerings and curriculum content within these programs from the university webpages. While similar efforts have been made to derive skills from US based educational programs [14, 15] no such efforts have been made for the Dutch context.

As such, our research aims to address the research questions: (1) What skills are encompassed within data science and AI master programs offered by Dutch universities? (2) What are the overall differences and similarities in the skills learned across different universities? (3)

How do the skills learned at more general data science and AI programs differ from those learned at more subject-specific data science master programs? In this context, we define 'general data science and AI programs' (G) as programs exclusively focused on this domain. We refer to 'subject-specific programs' (S) as those that concentrate on domain-specific applications of data science, such as data science for marketing. Moreover, it is important to clarify that our methodology employs unsupervised clustering combined with text mining techniques to derive a categorization of skills from the descriptions of the programs and courses. This approach ensures that the resulting clusters encapsulate a comprehensive range of knowledge, skills, and competencies (KSC), as characterized by [16]. These clusters are not intended to segregate the three dimensions of KSCs; rather, we use the term 'skills' as an inclusive term that embodies all three dimensions. Our objective is to provide a holistic understanding of the KSCs imparted by these master programs, thereby refraining from distinguishing between knowledge, skills, and competencies. For the sake of brevity and clarity in the remainder of this article, we will refer collectively to these elements as 'skills', implicitly encompassing the full spectrum of knowledge, skills, and competencies as discussed.

The remainder of this paper is organized as follows. In order to situate our study within the existing academic literature, we first review prior research on data science education and skill development. Next, we provide an overview of the text mining technique we employ, Correlated Topic Modeling (CTM), and discuss its potential for revealing insights into the diversity of skills taught in data science and AI study programs. Subsequently, we present the results. Finally, we will summarize our findings and discuss the implications for various stakeholders, including universities, businesses, and data science practitioners.

## 2. Literature review

### 2.1 Deriving data science skills from educational programs

Due to the ever growing interest in big data and analytics, the number of educational programs offering data driven skills has immensely grown globally in the last 10–15 years (Davenport, 2020). This growth has been driven by a combination of increasing market needs and theoretical development of the field of data science, as well as related fields such as business analytics or big data. Yet, the definition of the field is often driven by market needs and technology forecasts (for example [17]). Consequently, there is a wide variety in the skills taught across different data science programs [14, 15].

To our knowledge, at present only two studies have investigated the skills of data scientists by exploring the content of educational programs through text mining [14, 15]. First, [14] studied the content and structure of data science programs in higher education institutions in the United States. They conducted an exploratory content analysis of program descriptions, curriculum structures, and course focuses of 30 undergraduate and graduate programs in data science from various US based institutions. The study concluded that the investigated data science programs varied greatly in terms of specific technical skills, mathematical/statistical foundations, and domain-specific knowledge. However, core technical skills, such as programming, data management, and machine learning, were present in most programs. Finally, in all programs, only little emphasize on soft skills such as communication, collaboration, and ethical considerations was found.

Furthermore, [15] studied the content of academic curricula from undergraduate and post-graduate programs in data science and big data that was presented on the websites of 320 universities worldwide. Although they also observed a wide variety in the content of the different curricula, they came to the conclusion that core skills such as computer coding, statistics machine learning, visualisation and application innovation were core terms in most study programs.

Finally, [18] employed a somewhat different approach to investigate what skills should be taught in colleges of business to prepare students for entry-level careers in business analytics. To gain more insight in this, they studied the content of 23 US based undergraduate business analytics programs, employing a survey among 27 experts in data analytics, and content analysis of 215 job postings, requiring a 3 year college degree in business analytics and less than 3 years of professional experience. They concluded that while the findings of the three different study approaches differed slightly, there was a general agreement on the importance of business, analytical and technical skills for the entry-level careers in business analytics.

## 2.2 Towards a definition of the skills of data scientists

While there is still much ambiguity regarding the core skills that should be trained in data analytics study programs, interestingly enough, the last decade has seen a development of different organizations and agencies grown out of the need to define and regulate the understanding of the skills related to data driven jobs [2]. Such organizations are for example the Institute for Operations Research and management Science (INFORMS) Analytics Certification Job Task Analysis Working Group [2, 18] or the IADSS [2], the Edison project [10]. Also new journals have been started to propagate the development of the field of data science/ data driven thinking, for example the Harvard Data Science Review (started in 2019) or the Journal of Big Data (that received its first impact factor rating in 2022). Yet due to the rapid development of the field, society-level classifications accepted generally are not yet available [2] and most of the available classifications are U.S. based.

[10] offer a comprehensive review of current efforts to define the field of data science. They argue that arriving at a definitive description is premature due to the lack of consensus on its constitutive elements. However, they note that most definitions share a common emphasis on the integration of statistical and engineering skills. In the context of curriculum development, the authors reference seminal contributions by [19, 20]. [19] propose a framework for data science education in the United States, emphasizing four core components: big data infrastructure, big data analytics lifecycle, data management skills, and behavioural disciplines. Conversely, [2] and colleagues offer a comprehensive review of literature that aims to define the field of data science. They emphasize a range of essential competencies, including proficiency in big data tools, coding, statistical analysis, research, and data hacking. Additionally, they underscore the importance of ethical management and strategic thinking in data science.

In summary, there is a wide range of definitions and attempts on creating taxonomies available, that is specific to any scientific field that is not yet clearly defined or delimited. As also [10] highlight, common to all is a combination of statistical and engineering skills. This variety of definitions is specifically relevant from the perspective of this study, as our aim is to understand the skills of graduates of data science and AI programs that are coming from very different universities ranging from technical universities, to general research universities, but also specific programs in business schools or life science faculties. Given the diversity in the literature in defining core and domain specific fields, examining program curricula will help in understanding how this diversity is reflected in the Dutch educational system. We aim to investigate whether certain skills are common across graduate programs, thereby establishing a core set of skills, and to examine the variety of skills taught in programs with different focusses (e.g., general versus subject specific programs).

## 3. Methods

In order to answer the research questions, information is gathered about the content of all data science and AI master programs throughout the Netherlands for which data is openly

available. This information is gathered in the form of online descriptions of programs and courses. Thereafter, text analysis is performed using correlated topic modeling (CTM). Lastly, the posterior assignments obtained from the analysis have been used to highlight the inter-university differences.

The CTM is a hierarchical model of document collections that is developed from the Latent Dirichlet Allocation (LDA) based topic model [21] by allowing the topics to be correlated. LDA is a generative probabilistic model of clustering unstructured data. In this model, documents are represented as random mixtures over latent topics, where each topic is characterized by a distribution over words. Using such a latent variable based approach allows for a probabilistic classification of documents (in our case course and program descriptions) over multiple topics. The CTM models the words of each document from a mixture model. The components of this mixture model are shared by all documents in the collection, therefore the mixture proportions are document specific random variables. The CTM allows for multiple topics with different proportions for each document that are allowed to correlate. Thus, it allows to capture the heterogeneity in grouped data that show multiple latent patterns.

## 3.1 Data collection

The dataset was compiled by gathering program and course descriptions from all data science and AI master programs offered by Dutch research universities that have these descriptions available online. Consequently, the dataset includes 1,009 descriptions of courses and programs from a total of 41 master programs in data science and AI, which are offered by 11 universities and a joint program between 3 universities. Table 1 offers a comprehensive overview of the universities included in our study, including the number of programs and the core and elective courses analyzed. For one university, the official website did not distinguish between core and elective courses. To accommodate this, we have included a column in Table 1 that categorizes courses as core or elective based on their content. Additionally, S4 Appendix contains detailed information about the specific data science and AI master programs, their general or specific focus, and the demographics of the student body enrolled in each program.

**Table 1. Number of program and course descriptions per university included in the analysis.**

|  | Program | Core course | Elective | Core or elective |
|---|---|---|---|---|
| Delft University of Technology (TU Delft) | 1 | 10 | 66 | 0 |
| Eindhoven University of Technology (Tu/e) | 5 | 32 | 45 | 0 |
| Leiden University (LU) | 3 | 33 | 68 | 0 |
| Maastricht University (MU) | 5 | 26 | 76 | 0 |
| Radboud University Nijmegen (RAD) | 2 | 5 | 13 | 45 |
| Tilburg University (Til) | 5 | 39 | 64 | 0 |
| University of Groningen (RUG) | 2 | 22 | 29 | 0 |
| University of Twente (UT) | 6 | 34 | 87 | 0 |
| University of Amsterdam (UVA) | 5 | 49 | 37 | 0 |
| Utrecht University (UU) | 2 | 6 | 49 | 0 |
| Vrije Universiteit (VU) | 4 | 24 | 67 | 0 |
| Combination: Vrije Universiteit, Erasmus University of Rotterdam and Universiteit van Amsterdam[a] (Combi) | 1 | 20 | 22 | 0 |

*Note*. The number of program descriptions equals the number of programs selected for this study
[a]This is a program offered by three different universities and is thus a combination of institutions

**Table 2. Manual changes made to the texts as part of data pre-processing.**

| Change | Words |
|---|---|
| Plural to singular | models, systems, sets, problems, networks, games |
| Words combined | machine learning, deep learning, statistical learning, data science, computer science, artificial intelligence, data mining, text mining, time series, neural network, research project, distributed computing, natural language processing, probabilistic theory, distributed system, critical thinking, decision making, skills, ad hoc |
| Words deleted | will, course, courses, student, students, able, university, master, can, skills, work, new, use, used, using, also, different, learn, learning, part, master's, understand, one, two, game, topics, understanding, based, many, several, exam, make, discussed, ad hoc |

## 3.2 Data processing

After collecting the descriptions, some altercations have been made to the text in order to obtain more optimized results. The specific changes are stated in Table 2. The text analyses is based on a count of words, defined as "Term Frequency". To optimize the results we combined manually words that occur frequently and have a special meaning in the combined form, for example: "statistical_learning" or "artificial_inteligence". Furthermore some plural forms have been changed into singular form. Words that did not hold any information and numbers in general have been removed and all letters have been set to lower case.

## 4. Results

Fig 1 presents the coherence scores for CTM with 2 to 30 clusters. While the optimum is at 13 topics, at 7 topics the graph shows a clear elbow, as such we will investigate both the 7 and 13 cluster solution.

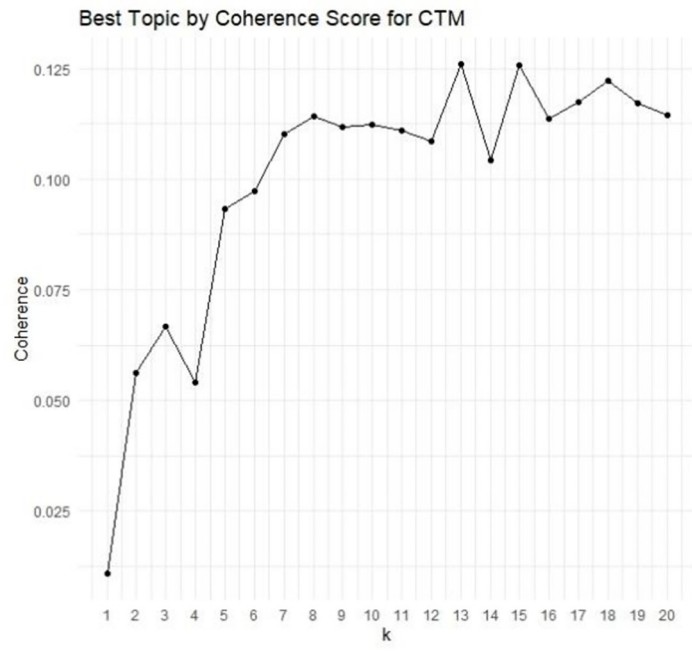

**Fig 1. Plot of the coherence scores per number of K topics for the CTM.**

In Table 3 the results for 7 topics are shown. We see a large number of subdomains and corresponding skills being described already by the 7 correlated clusters. Apart from the first topic describing core terms, every topic denotes a domain within data science and AI meaning skills of this field are being expounded in the descriptions. Interestingly enough the sixth topic is about ethics which is a less often discussed subdomain yet with a growing societal relevance.

When compared to the CTM model with 13 topics (see S3 Appendix), we see there is a lot of overlap with the CTM model with 7 topics. Main differences include ethics not being denoted in 13 cluster model and more subdomains being described in the more complex model such as machine learning, natural language processing, AI/optimization, the human brain, information security and the business domain. Since the CTM model with 7 topics gives a sufficient overview of the corpus, this model has been chosen as the main model.

When we look at the indicated clusters of skills in Table 3, we can clearly see that there is a wide variety of skills needed to complete a data science master program in the Netherlands. At the same time, there is also some overlap between the 7 topics. Most predominantly, the word "data" is present in every topic, not surprisingly, given its central role in these master programs. Also the words "model" (present in topics 3, 4, 5, 7), "Methods" (topics 3, 4, 7) and "machine learning" (topics 3, 4) are central to the clustering. Their presence in all these different clusters shows the wide spread importance of these skills. Each topic, however, has a specific focus characterized by the composition of the most frequent 10 words. For example, topic 2 is described by words like "statistical", "theory", "linear", "data analysis"- that relate more to the more traditional statistical aspects of data science. In contrast, topic 3 focuses on "deep learning", "natural language processing", "machine learning"- more innovative elements of a broader data science field, while topic 5 encompasses skills like "health", "ethical", "decision", pointing to applications of data science in different domains. We manually labelled each topic based on the 10 most frequent words and distinguished between core elements of data science and subdomains in Table 4 to aid interpretation.

Skills that correspond to the mentioned fields in Table 4 are taught in data science related master programs throughout the Netherlands. Based on these results, it might be useful for employers to investigate these terms closer to get a better grasp on what data science graduates have to offer, in order to match the skills to the concerning vacancies. However, there is quite a variability between universities in the core and subdomain skills they prioritize. In order to have a grasp on this diversity we continue with switching our focus to universities.

We classified the universities based on the most prominent topics in their course and program descriptions. The classification is performed by taking the mean of the posteriors of all

**Table 3. Top ten most common words per topic for the CTM where K = 7 with the topic label in last row.**

| | Topic 1 | Topic 2 | Topic 3 | Topic 4 | Topic 5 | Topic 6 | Topic 7 |
|---|---|---|---|---|---|---|---|
| 1 | research | model | data | algorithms | data | system | research |
| 2 | project | Data | model | machinelearning | system | artificialintelligence | model |
| 3 | datascience | analysis | techniques | data | information | data | social |
| 4 | business | techniques | process | techniques | software | health | system |
| 5 | data | image | deeplearning | methods | business | concepts | knowledge |
| 6 | programme | statistical | naturallanguageprocessing | model | design | datascience | design |
| 7 | knowledge | theory | language | knowledge | services | ethical | network |
| 8 | scientific | methods | machinelearning | programming | web | decisions | data |
| 9 | development | linear | methods | problem | model | privacy | methods |
| 10 | thesis | computer | datascience | datamining | distributed | problem | human |
| Label | *Core elements of data science* | *Statistics* | *Data processing techniques 3* | *Data processing techniques 4* | *Electronics-IT* | *Ethics* | *Research* |

**Table 4. Sorted skills based on manually named topics according to CTM.**

| *Data science specific skills* | *Skills of subdomains* |
|---|---|
| Network (analysis) | Law |
| Deep learning | Health |
| Data mining | Research |
| Natural language processing | Business/marketing |
| Artificial intelligence | Ethics |
| Machine learning | Research |
| Data analysing | |
| IT | |
| Statistics | |
| Data processing techniques | |

descriptions per university per topic averaged across the programs. As such, this profile describes not the whole university, just the investigated programs.

We see in Table 5 that there are a lot of differences in order between the general outcome and the individual universities. Core terms, research and ethics are very prominent topics for multiple universities. A small cluster consisting of Rad, Til, RUG and UU that all share the first two topics, namely research and ethics, is also uncovered. However, apart from these two corresponding topics, the order of the topics that come after differ a lot. Lastly, the two technical universities both score high on electronics/IT, which is the more technical topic.

**Table 5. Order of most important manually labelled topic terms per university based on posterior values for the CTM.**

| | University | First | Second | Third | Fourth | Fifth | Sixth | Seventh |
|---|---|---|---|---|---|---|---|---|
| 1 | TU Delft | Electronics/IT | Data processing techniques_4 | Research | Statistics | Ethics | Data processing techniques_3 | Core terms |
| 2 | TU/e | Core terms | Electronics/IT | Research | Data processing techniques_4 | Ethics | Statistics | Data processing techniques_3 |
| 3 | LU | Core terms | Data processing techniques_4 | Research | Statistics | Electronics/IT | Data processing techniques_3 | Ethics |
| 4 | MU | Data processing techniques_4 | Data processing techniques_3 | Ethics | Statistics | Core terms | Research | Electronics/IT |
| 5 | RAD | Research | Ethics | Data processing techniques_4 | Core terms | Data processing techniques_3 | Statistics | Electronics/IT |
| 6 | Til | Research | Ethics | Core terms | Statistics | Electronics/IT | Data processing techniques_3 | Data processing techniques_4 |
| 7 | RUG | Research | Ethics | Core terms | Data processing techniques_3 | Statistics | Data processing techniques_4 | Electronics/IT |
| 8 | UT | Data processing techniques_3 | Electronics/IT | Statistics | Ethics | Research | Core terms | Data processing techniques_4 |
| 9 | UvA | Core terms | Research | Ethics | Data processing techniques_4 | Statistics | Data processing techniques_3 | Electronics/IT |
| 10 | UU | Research | Ethics | Statistics | Data processing techniques_4 | Core terms | Data processing techniques_3 | Electronics/IT |
| 11 | VU | Research | Core terms | Data processing techniques_4 | Statistics | Data processing techniques_3 | Electronics/IT | Ethics |
| 12 | Combi * | Statistics | Research | Core terms | Data processing techniques_4 | Ethics | Electronics/IT | Data processing techniques_3 |

\* Combi is defined as one program offered by three universities simultaneously UvA, VU and Erasmus university Rotterdam. The second line represents a manually added key term describing the main topics per cluster

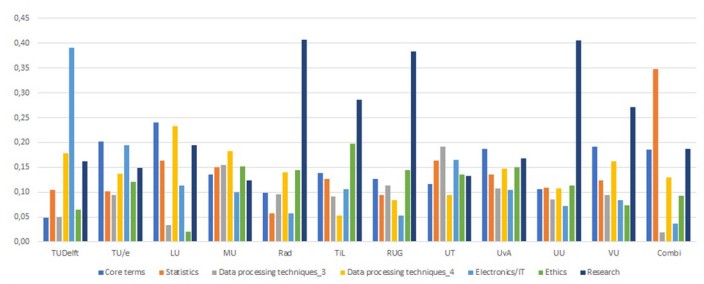

**Fig 2. Seven bar plots of the posteriors per university for every manually labelled topic of the CTM outcomes.**

In Fig 2, we switch the lenses for the presentation to show the prevalence of the seven topics across the universities. These visualisations are plotted to help identify which universities are similar in scoring high on a given topic. Note that when universities score high on a certain topic, it means that their data science and AI programs are more oriented towards the described topic. It does not mean that the entire university is oriented towards the concerning domain. The bar chart clearly visualizes the diversity among universities with regard to the prevalence of the different topics.

TU Delft stands out by the relative low prevalence of core terms coupled with the highest prevalence of the Electronics/ IT related topic. On IT/electronics unsurprisingly the second highest score is for Tu/E. The prevalence of the second cluster, focusing on statistics seems to be very balanced across universities, with the Combi program being the only one that focuses more heavily on this pillar. On the research cluster, the more general universities (RAD, RIU, UU) score higher than the technical universities. The differences between the two data processing clusters are also large between universities, with for example TU Delft, TU/e and LU having more focus on the second data processing pillar, while UT on the 1st, with MU, Uva, VU, and TiL a more balanced distribution between the pillars. As such, industries with special interest of skills corresponding to certain pillars can be well informed by such a distribution of focus between programs. LU scores specifically low on the Ethics dimension- but we need to mention that for this university only the more technical programs were selected (not including the law faculty).

Even within the same university, there can be large differences between skills taught, depending on the focus of the master programs. For illustrative purposes, S5 Appendix presents the posterior distribution of the seven topics across the two master programs offered by Utrecht and Groningen University. At Utrecht University, both the Applied Data Science and Artificial Intelligence programs have a similar focus on the research pillar. However, the Applied Data Science program has a heavier focus on core terms and statistics than the Artificial Intelligence program. At the same time, at Groningen University, the specialized Marketing Analytics and Data Science program shows less emphasis on the second to forth pillars than the general Artificial Intelligence program. Interestingly, the two AI master programs at these universities are more similar to each other than to the two more applied, specialized programs, and this pattern is also observed in the subject specific programs. In S6 Appendix, we present the posterior distribution of each topic per course for the four programs. Although this table may be overly detailed for analytical purposes, it can be very informative for prospective master students and employers. The detailed distribution of the seven topics for all courses is available in the R environment in the OSF file DOI [osf.io/64xf7].

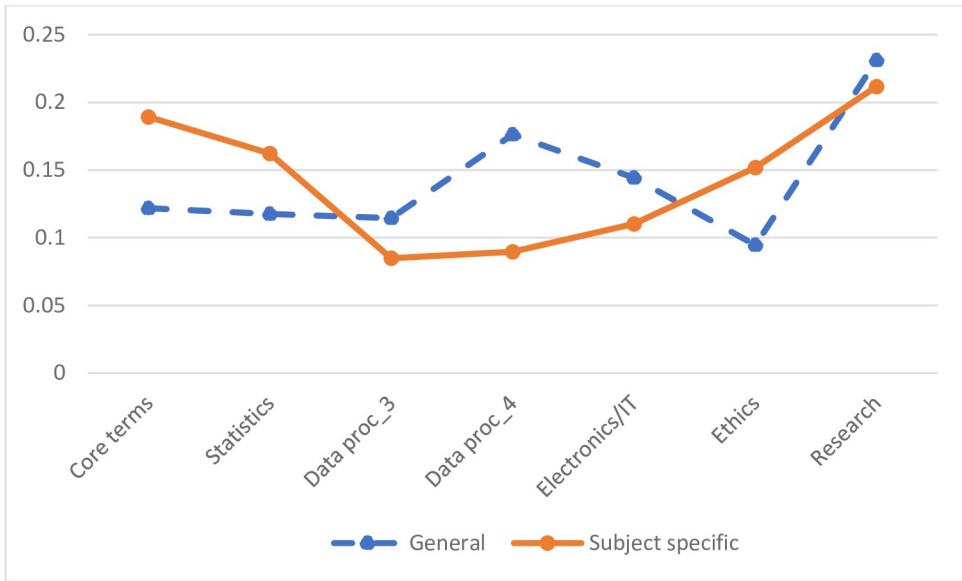

**Fig 3. The posterior distribution per program type (general vs subject specific) of the 7 topics of the CTM model.**
The classification of programs into general vs. subject specific are available in S4 Appendix.

To address Research Question 3, we examined the distribution of posterior classifications by program type, as depicted in Fig 3. This analysis distinguishes between general programs—with a broad focus on data science and AI skills—and subject-specific programs, which concentrate on a more narrow, domain-specific application of data science, such as in life sciences or marketing. Similar to the more detailed comparison in S5 Appendix, Fig 3 also shows a marked difference between general programs, which focus on the more technical pillars 3–5, and the subject-specific programs, which stand out in the first two, and last research pillars, thus the less technical pillars.

Fig 3 illustrates a pronounced distinction between the two types of programs. General programs tend to score higher in the data processing clusters, particularly in research and electronics/IT. Conversely, subject-specific programs predominantly feature core terms related to statistics and ethics. A closer examination of the defining words within these clusters reveals that subject-specific programs not only emphasize data science technical skills but also integrate terms associated with ethics, health, and business. In contrast, general programs are characterized by a focus on technical and engineering skills.

## 5. Discussion and conclusion

In this article, we set out to examine the skills of data science and AI graduates in the Netherlands. Our goal was to gain a better understanding of what skills are taught within data science and AI master programs offered by Dutch universities and how they differ or are similar across institutions. Additionally, we wanted to compare the skills learned in more general data science and AI programs with those learned in subject-specific data science and AI master programs. To achieve this, we employed a text mining technique called CTM (Correlated Topic Modeling) on 1009 program and course descriptions from data science and AI master programs of 11 Dutch universities and a combined interuniversity program between 3 universities. This technique allowed us to extract structured latent variables from the program and course descriptions and to model the semantic relationships between topics and words.

The results indicate that the majority of the Dutch universities teach core skills such as research, ethics, and core terms, which likely form the foundation of data science and AI education. These findings are in line with the results of other studies that have emphasized the importance of ethics, research skills, and technical skills in data science education [10, 11, 14, 22]. Moreover, the results of our study reveal that while the prevalence of these topics is relatively consistent across universities, the order of the topics and the specific skills taught can vary significantly.

The results also suggest that Dutch data science and AI programs are highly diverse, both in terms of the skills taught and the focus of the programs. Specifically, the technical universities, such as TU Delft and TU/e, excel in electronics and IT, while the more comprehensive universities demonstrate a stronger emphasis on research. Moreover, the results show that while general data science and AI programs typically focus on data processing, electronics/IT, and research, subject-specific programs place a greater emphasis on statistics and ethics. These findings are in line with the results of prior research that has noted a wide range of skills taught across different data science programs [14, 15]. Our study provides a more nuanced understanding of this diversity, by showing that the diversity in the skills taught is not only dependent on the type of university, but also on the specific programs offered within each university.

A notable finding from our analysis is furthermore the absence of soft skills in the clustering results. Neither the 7-topic nor the more comprehensive 13-topic solutions revealed any reference to soft skills within the curricula. This suggests that soft skills are not a central or frequent component in the master programs we examined. While this finding aligns with previous research, which also highlights a predominant focus on technical skills [14, 15, 18], it remains a concerning trend, especially considering research that highlights the demand for soft skills in the job market [11, 12, 23]. In light of this discrepancy between curricular content and market needs, further investigation, possibly through qualitative methods such as interviews with program managers, is recommended to explore the reasons behind this lack of emphasis on soft skills.

The results of our study have important implications for various stakeholders. For universities and other educational institutions, our findings provide valuable insights into the content of data science and AI master programs and can be used to refine and update curricula. For businesses and organizations, the results can help to better match the qualifications of candidates to their specific needs. Finally, for current and aspiring data scientists, our findings can be used to identify gaps in their own skills and guide their professional development efforts.

However, it is important to note that our study has several limitations. Firstly, our results are based on the analysis of university program websites and may not reflect the actual skills taught in the classroom. Naturally, specific skills cannot be derived from the program texts on university website alone. As a result, it is hard to derive specific information from the text analyses without further knowledge of this field of science. Therefore, expert knowledge will be needed to fully understand what data science and AI graduates are capable of doing. Additional attention will have to be put in by further research to get a good grasp on what the domains of data science and AI exactly consist of.

Furthermore, both program and course descriptions have been used in these analyses. This leads to very broad descriptions and more narrowed down descriptions being put on the same pile and being considered as the same time. In addition, some programs have a higher number of courses than other programs, which might cause the results to be skewed towards the programs with a higher number of courses (and thus a higher number of descriptions and therefore a higher number of words). These programs might have more influence on the results than programs with a lower number of courses or less detailed descriptions. This is also relevant for the posterior classification, since different universities have different totals of

programs included in this study. Therefore, results might be skewed towards the universities with a higher number of programs included with more detailed descriptions.

Another shortcoming of our approach is that although the data has a multilevel structure, with courses and programs nested within universities, we did not explicitly model this multi-level structure. The reason for this is that currently no topic modelling approach is available that can properly model the hierarchical structure in the data.

Despite these shortcomings, the findings of the present study do shed light on the very ill defined concept of data science and AI. By examining 1,009 master program and course descriptions, the most important aspects of data science and AI were uncovered, marking the beginning of a larger effort to fully define the knowledge domain of these graduates. This study fills a significant gap in literature by specifically focusing on the skills of these graduates and can be used to better define the emerging field from a practical perspective, as well as to bridge the gap between the supply of education and the demand of the job market. Further research could explore the skills that graduates actually use in their first jobs, and how this relates to their studies. Additionally, investigating the connection between the needs of specific market segments and the profiles of different universities could help to improve educational programs and assist students in making informed decisions about their future employment prospects.

## Supporting information

**S1 Appendix. Search keys data science and artificial intelligence.**
(DOCX)

**S2 Appendix. Erasing non-informative words: "ad" and "hoc".**
(DOCX)

**S3 Appendix. Top ten most frequent words per topic for the CTM with 13 topics.**
(DOCX)

**S4 Appendix. Overview of 2021 master programs: Distribution by institution and corpus of student body by enrollment numbers, categorized into general (G) or specific (S) programs.**
(DOCX)

**S5 Appendix. The posterior distribution of the 7 topics across the 4 master programs offered at Utrecht (UU, top) and Groningen University (RUG, bottom).**
(DOCX)

**S6 Appendix. Posterior classification over the 7 topics of the courses offered in the 4 master programs at Utrecht University (UU) and Groningen University (RUG).**
(DOCX)

## Acknowledgments

We would like to thank Adham Kahlawi for insightful feedback about the data analysis and data quality on an initial draft of the ms.

## Author Contributions

**Conceptualization:** Zsuzsa Bakk.

**Data curation:** Mathijs J. Mol.

**Formal analysis:** Mathijs J. Mol.

**Methodology:** Mathijs J. Mol, Zsuzsa Bakk.

**Resources:** Barbara Belfi.

**Supervision:** Zsuzsa Bakk.

**Writing – original draft:** Mathijs J. Mol.

**Writing – review & editing:** Barbara Belfi, Zsuzsa Bakk.

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
