## [Decision Letter · Decision Letter 0]

15 Nov 2023

PONE-D-23-16749Unraveling the Skillsets of Data Scientists:  Text Mining Analysis of Dutch University Master Programs in Data Science and Artificial IntelligencePLOS ONE

Dear Dr. Bakk,

Thank you for submitting your manuscript to PLOS ONE. After careful consideration, we feel that it has merit but does not fully meet PLOS ONE’s publication criteria as it currently stands. Therefore, we invite you to submit a revised version of the manuscript that addresses the points raised during the review process.

Dear Authors,

Subject: Manuscript Decision – Major Revision Required

Following a thorough evaluation of your manuscript, which applies an innovative methodology to study the skillset of data science Master's students in Dutch higher education institutions, the decision is that a major revision is required prior to considering its publication. Your article presents a timely and insightful contribution, with quality writing and structure, referencing relevant literature, and providing insightful findings with practical implications. However, I have identified several issues that need to be addressed to enhance the clarity and coherence of the work.

1) Inconsistency in the Number of Institutions: On pages 2 and 15, 7 Dutch universities are mentioned, while on page 10, 11 institutions and a collaboration of 3 institutions are referred to. It is important to clarify the exact number of higher education institutions included in the sample and ensure this information is consistent throughout the article.

2) Clarification of the Third Research Question (RQ3): I request a clarification on what is meant by "general" vs. "subject-specific" data science programs. Providing examples or further explanation of this distinction would be helpful.

3) Use of Terms "Skills", "Competencies", "Knowledge": These terms are interchangeably used but do not exactly mean the same thing. I recommend choosing one term to use consistently and defining it clearly.

4) Inclusion of Institutions from the VSNU Group: On page 10, it is mentioned that only institutions that are part of the VSNU group were included for reliability reasons. Please explain what this means, how this group compares to other higher education institutions, and how representative it is of the educational system.

5) Addition of Information in the Annex: It would be beneficial to include a table in the Annex listing the exact names/titles of the data science programs per institution. Additionally, understanding the size of the corpus for each institution would be useful.

6)*Inclusion of "Soft Skills": I would like to know what your findings suggest about the inclusion of "soft skills" in Dutch data science Master's programs.

Please consider these points and make the necessary revisions. I am available for any further clarifications that may be required during the revision process.

Sincerely,

We look forward to receiving your revised manuscript.

Kind regards,

Ricardo Limongi

Academic Editor

PLOS ONE

Additional Editor Comments :

Dear Authors,

Following a thorough evaluation of your manuscript, which applies an innovative methodology to study the skillset of data science Master's students in Dutch higher education institutions, the decision is that a major revision is required prior to considering its publication. Your article presents a timely and insightful contribution, with quality writing and structure, referencing relevant literature, and providing insightful findings with practical implications. However, I have identified several issues that need to be addressed to enhance the clarity and coherence of the work.

1) Inconsistency in the Number of Institutions: On pages 2 and 15, 7 Dutch universities are mentioned, while on page 10, 11 institutions and a collaboration of 3 institutions are referred to. It is important to clarify the exact number of higher education institutions included in the sample and ensure this information is consistent throughout the article.

2) Clarification of the Third Research Question (RQ3): I request a clarification on what is meant by "general" vs. "subject-specific" data science programs. Providing examples or further explanation of this distinction would be helpful.

3) Use of Terms "Skills", "Competencies", "Knowledge": These terms are interchangeably used but do not exactly mean the same thing. I recommend choosing one term to use consistently and defining it clearly.

4) Inclusion of Institutions from the VSNU Group: On page 10, it is mentioned that only institutions that are part of the VSNU group were included for reliability reasons. Please explain what this means, how this group compares to other higher education institutions, and how representative it is of the educational system.

5) Addition of Information in the Annex: It would be beneficial to include a table in the Annex listing the exact names/titles of the data science programs per institution. Additionally, understanding the size of the corpus for each institution would be useful.

6) Inclusion of "Soft Skills": I would like to know what your findings suggest about the inclusion of "soft skills" in Dutch data science Master's programs.

Please consider these points and make the necessary revisions. I am available for any further clarifications that may be required during the revision process.

Sincerely,

Reviewers' comments:

Reviewer's Responses to Questions

**Comments to the Author**

1. Is the manuscript technically sound, and do the data support the conclusions?

Reviewer #1: Partly

Reviewer #2: Yes

2. Has the statistical analysis been performed appropriately and rigorously? 

Reviewer #1: Yes

Reviewer #2: Yes

3. Have the authors made all data underlying the findings in their manuscript fully available?

Reviewer #1: Yes

Reviewer #2: Yes

4. Is the manuscript presented in an intelligible fashion and written in standard English?

Reviewer #1: Yes

Reviewer #2: Yes

5. Review Comments to the Author

Reviewer #1: The paper presents similarities and differences across 41 master programs in data science and artificial intelligence in dutch universities. The dataset is relatively small (1009 courses from 14 universities). The results were obtained through text mining.

The paper is generally well written and organized. The method is presented in a way that is concise yet easy to follow. In my opinion, the approach of applying text mining to discuss differences and similarities in collections of documents are far more interesting than the particular results found in this paper (data science programs are different; so what?).

In my opinion the paper has a promising contribution that needs to be strengthened with the following:

- I would like to challenge the following statement about the growth in data science programs: …“has mostly been driven by market needs, and is not preceded by a theoretical development of the field of data science”. I would argue the opposite. Theoretical achievements in the past decade such as large language models and generative adversarial networks were a key driver for the revival of research and educational programs in data science. I am afraid other readers of the paper may rate this work negatively if this issue is not addressed more thoroughly.

- The results need to be explained in more detail to support conclusions. For example, the word “data” is present in all 7 topics. Does it mean that “data” is irrelevant to discriminate and classify courses among topics? As another example, posteriors and their means lack introduction (context), explanation and discussion as the relevance for the distribution of topics for each university.

- While the discussion focuses on universities and topics, there is no discussion around courses, which seem to define the unit of analysis in the data (i.e. each of the 1009 examples is a course). Does CTM enable analysis of “within group” variability, i.e. are courses and programs within a university more similar than comparing courses between different universities? The hierarchical clustering (i.e. courses within programs within universities) needs to be considered in the analysis somehow.

- Figures (and their fonts) must be resized for readability.

- Spell checking and a grammar review are required. One example of typo is in the first paragraph of section 3: “corelated.”

Reviewer #2: This is an insightful and timely contribution, which applies a novel methodology to study the skillset of data science Master's students in Dutch higher education institutions. The article is well-written, well-structured, references relevant literature on the topic and provides insightful findings with implications for practice. I would make a few minor recommendations that would further improve its clarity and coherence. Please find my suggestions below (in no particular order):

1) On p. 2 in the "Abstract" and p. 15 "Discussion and Conclusion" the authors mention the inclusion of 7 Dutch universities, while p. 10 in "Data collection" section references 11 institutions and a collaboration of 3 institutions. Please clarify how many higher education institutions you included in the sample and ensure that this is coherent throughout the paper.

2) I would recommend to clarify your 3rd RQ. What exactly do you mean with "general" vs. "subject-specific" data science programs? It would be good to provide some examples here or explain further.

3) The authors use the terms "skills", "competencies" "knowledge" interchangeably throughout the paper, while they do not exactly mean the same thing. I would recommend, for clarity reasons, to stick to one term and define it.

4) On p. 10 you mention that you included only institutions that are part of the VSNU group, for reliability reasons. What exactly is meant here? How does this group compare to other higher education institutions (i.e. what proportion of the system is represented in your sample)? How representative is it?

5) It would be informative to add in the Annex a table that lists the exact names/titles of the programs in data science per institution. With that, I would also find it beneficial to understand the size of the corpora for each institution.

6) What do your findings suggest about the inclusion of "soft skills" in Dutch data science Master's programs?

6. PLOS authors have the option to publish the peer review history of their article (what does this mean?). If published, this will include your full peer review and any attached files.

Reviewer #1: No

Reviewer #2: No

---

## [Author Response · Author response to Decision Letter 0]

27 Nov 2023

Dear Editor,

We would like to thank you and the reviewers for the very useful comments. I will show below in detail after each comment how we addressed the comment in the manuscript using italic, and provide citations from the revised manuscript where relevant. 

Kind regards,

The authors

Dear Authors,

Subject: Manuscript Decision – Major Revision Required

Following a thorough evaluation of your manuscript, which applies an innovative methodology to study the skillset of data science Master's students in Dutch higher education institutions, the decision is that a major revision is required prior to considering its publication. Your article presents a timely and insightful contribution, with quality writing and structure, referencing relevant literature, and providing insightful findings with practical implications. However, I have identified several issues that need to be addressed to enhance the clarity and coherence of the work.

1) Inconsistency in the Number of Institutions: On pages 2 and 15, 7 Dutch universities are mentioned, while on page 10, 11 institutions and a collaboration of 3 institutions are referred to. It is important to clarify the exact number of higher education institutions included in the sample and ensure this information is consistent throughout the article.

We would like to thank you for pointing out this inconsistency. Indeed we have 11 universities in the sample 1 plus an MSc program that is organized in cooperation between 3 universities as a joint program. We now corrected this error in the abstract and on page 15. 

2) Clarification of the Third Research Question (RQ3): I request a clarification on what is meant by "general" vs. "subject-specific" data science programs. Providing examples or further explanation of this distinction would be helpful.

Thank you for this comments. Indeed the differentiation was not clearly specified- we meant by general programs data science programs that are not field specific, and by subject-specific programs, programs that focus on data science within a disciplinary context for example Data science for marketing, or for the social sciences. – we now add an explanation on page 4 (see also citation below), and in the new appendix D in the last column we provide the classification for each master program. 

“(3) How do the skillsets learned at more general data science and AI programs differ from those learned at more subject-specific data science master programs? In this context, we define 'general data science and AI programs' (G) as programs exclusively focused on this domain. We refer to 'subject-specific programs' (S) as those that concentrate on domain-specific applications of data science, such as data science for marketing.” 

Furthermore, we also discuss this in more detail in the results section, where the new Figure 3 zooms in more specifically on the distribution of the 7 topics over these two program types, see page 13: 

“To address Research Question 3, we examined the distribution of posterior classifications by program type, as depicted in Figure 3, distinguishing between general programs—those with a broad focus on data science and AI skills—and subject-specific programs, which concentrate on a more narrow, domain-specific application of data science, such as in life sciences or marketing. 

[Figure 3 about here]

Figure 3 illustrates a pronounced distinction between the two types of programs. General programs tend to score higher in the data processing clusters, particularly in research and electronics/IT. Conversely, subject-specific programs predominantly feature core terms related to statistics and ethics. A closer examination of the defining words within these clusters reveals that subject-specific programs not only emphasize data science technical skills but also integrate terms associated with ethics, health, and business. In contrast, general programs are characterized by a focus on technical and engineering skill sets.”

Finally, we now also refer more explicitly to the findings of Research Question 3 in the abstract and discussion and conclusion section of the article.

3) Use of Terms "Skills", "Competencies", "Knowledge": These terms are interchangeably used but do not exactly mean the same thing. I recommend choosing one term to use consistently and defining it clearly.

We agree with the reviewer that we used these terms loosely. The reason for this is that because we extracted the key terms directly from the data using unsupervised clustering some of the key terms are skills, while others fall under competencies or knowledge domains. We explicitly explain this issue in the current version on page 4: 

“In this context, we define 'general data science and AI programs' (G) as programs exclusively focused on this domain. We refer to 'subject-specific programs' (S) as those that concentrate on domain-specific applications of data science, such as data science for marketing. Moreover, it is important to clarify that our methodology employs unsupervised clustering combined with text mining techniques to derive a categorization of skills from the descriptions of the programs and courses. This approach ensures that the resulting clusters encapsulate a comprehensive range of knowledge, skills, and competencies (KSC), as characterized by Heller, Unlu, and Albert (2013). Our objective is to provide a holistic understanding of the KSCs imparted by these master's programs, thereby refraining from distinguishing between KSCs. For the sake of brevity and clarity in the remainder of this article, we will refer collectively to these elements as 'skills', implicitly encompassing the full spectrum of KSCs as discussed.”

We use “skills” thereafter as the only general term. The term skill set is used when referred to a set of skills.

4) Inclusion of Institutions from the VSNU Group: On page 10, it is mentioned that only institutions that are part of the VSNU group were included for reliability reasons. Please explain what this means, how this group compares to other higher education institutions, and how representative it is of the educational system.

We appreciate the reviewer's request for clarification regarding the inclusion of institutions from the VSNU group. The VSNU, now known as 'The Universities of the Netherlands', is a consortium recognized by the Dutch government that comprises all major government-funded universities in the Netherlands. This includes ten research universities, three specialized institutions, and one open university, which collectively represent 74% of the country's research universities. Membership in the VSNU assures adherence to stringent quality benchmarks and uniform academic standards across these institutions.

The excluded 26% of research universities consist of five small, specialized institutions: three theological universities dedicated to theological studies for clergy, one military university focused on defense education, and one private business school. Notably, these institutions do not offer master's programs in data science or artificial intelligence, which are the focus of our study.

To maintain the focus on the research scope and to avoid potential confusion, we have omitted the detailed information about the VSNU in our revised manuscript. We have instead emphasized that our analysis encompasses all Dutch master's programs in data science and AI for which course and/or program descriptions were publicly available, ensuring the reliability of our results. This refinement clarifies that the study is comprehensive within the context of the relevant educational programs in the Netherlands.

5) Addition of Information in the Annex: It would be beneficial to include a table in the Annex listing the exact names/titles of the data science programs per institution. Additionally, understanding the size of the corpus for each institution would be useful.

We added an extra table with this info into the new appendix D that shows all programs per university and the size of the corpus for the academic year 2020/21, the year of data collection. This table also has a column, that shows if a program was classified as general Data science/ AI (G) or domain specific (S) that helps interpreting the results for RQ3. 

6)*Inclusion of "Soft Skills": I would like to know what your findings suggest about the inclusion of "soft skills" in Dutch data science Master's programs.

Thank you for bringing this up, to motivate us to further elaborate. We actually found it concerning how using the text mining approach soft skills did not surface at all among the skills that define the clustering. We did a data driven approach- so this means this skills are not central enough/ not present to inform the clustering. We suggest for follow up research to look into more details about why these skills are not central. Follow up research should look into this very important issue in more detail, using different approaches- for example by interviewing program coordinators, because the course descriptions in their own did not provide sufficient data on this very important topic. We specifically discuss the problem in the discussion on page 14:

“A notable finding from our analysis is furthermore the absence of soft skills in the clustering results. Neither the 7-topic nor the more comprehensive 13-topic solutions revealed any reference to soft skills within the curricula. This observation underscores that soft skills are not a central or frequent component in the master's programs we examined. While this finding aligns with previous research, which also highlights a predominant focus on technical skills (Tang and Sae-Lim, 2016; West, 2017; Cegielksi and Jones-Farmer, 2016), it remains a concerning trend. Nonetheless, this trend is concerning, particularly when juxtaposed with research highlighting the demand for soft skills in the job market, underscoring their significance (Allen et al., 2021; Mauro et al., 2017; Markow et al., 2017). In light of this discrepancy between curricular content and market needs, we recommend further investigation, possibly through qualitative methods such as interviews with program managers, to explore the reasons behind this lack of emphasis on soft skills.”

Please consider these points and make the necessary revisions. I am available for any further clarifications that may be required during the revision process.

Sincerely,

We look forward to receiving your revised manuscript.

Kind regards,

Ricardo Limongi

Academic Editor

PLOS ONE

---

## [Editor Report · Decision Letter 1]

12 Dec 2023

PONE-D-23-16749R1Unravelling the Skill Sets of Data Scientists:  A Text Mining Analysis of Dutch University Master Programs in Data Science and Artificial IntelligencePLOS ONE

Dear Dr. Bakk,

Thank you for submitting your manuscript to PLOS ONE. After careful consideration, we feel that it has merit but does not fully meet PLOS ONE’s publication criteria as it currently stands. Therefore, we invite you to submit a revised version of the manuscript that addresses the points raised during the review process.

We look forward to receiving your revised manuscript.

Kind regards,

Ricardo Limongi

Academic Editor

PLOS ONE

Journal Requirements:

Additional Editor Comments:

Dear Authors,

Thank you for submitting your revised manuscript to our journal. After a thorough evaluation by our reviewers, it has been determined that additional revisions are necessary to enhance the quality and relevance of your work. The topic you are addressing is indeed intriguing and holds potential significance in the field. However, to ensure that your paper makes a substantial contribution to the literature, the following improvements are essential:

1. Network Structure and Transfer Analysis: The reviewers have highlighted the importance of including transfers from the second division, especially from B-teams of championship contender clubs to mid-table clubs. This aspect forms a crucial part of the network structure in your study area and should be incorporated into your analysis to provide a more comprehensive understanding of soccer transfers.

2. Incorporation of Sporting Variables: Considering that your paper focuses on sports, specifically soccer, it is critical to include analysis of sporting variables that influence team playing styles. This addition will enhance the relevance of your study by linking financial aspects of player transfers with the qualitative improvement of teams.

3. Clarification of Knowledge Gap: There is a need for a clearer articulation of the knowledge gap your study aims to fill. This will provide a stronger foundation for your research and its significance within the existing literature.

4. Theoretical Foundation and Hypothesis Support: The introduction requires a more robust theoretical framework. Additionally, the hypotheses presented must be substantiated with adequate theoretical support to strengthen the overall argument of your paper.

5. Updating the Study Sample: The current study sample, based on the 2017/2018 and 2018/2019 seasons, is dated. Updating the sample to include more recent seasons will enhance the relevance and applicability of your findings.

6. Enhanced Discussion Section: The "Empirical Analysis" section of your paper needs a deeper and more critical discussion. This should involve a comparative analysis with other authors' works in the field to contextualize your findings within the broader academic discourse.

We appreciate your efforts in conducting this research and encourage you to undertake these revisions to bring your manuscript to the requisite standard for publication. Your revised manuscript will undergo another round of review. Please feel free to reach out if you need any further guidance or clarification.

We look forward to receiving your revised submission.

Sincerely,

---

## [Author Response · Author response to Decision Letter 1]

2 Feb 2024

Dear Zuzsa Bakk,

Recommendations for Revisions - Unravelling the Skill Sets of Data Scientists: A Text Mining Analysis of Dutch University Master Programs in Data Science and Artificial Intelligence

Your manuscript, examining similarities and differences across master programs in data science and artificial intelligence in Dutch universities, presents an insightful and timely contribution to the field. The novel methodology applied to study the skillset of data science Master's students in Dutch higher education institutions is commendable. The manuscript is well-written and well-structured, with relevant literature references and insightful findings. However, to strengthen the manuscript for potential publication, I recommend addressing the following points:

1. The assertion that the growth in data science programs has been driven more by market needs than theoretical development warrants a more nuanced discussion. Consider incorporating viewpoints that acknowledge the significant role of theoretical advancements in the field, such as large language models and generative adversarial networks, in driving the evolution of research and educational programs in data science.

We would like to thank the reviewer for highlighting this point. We agree that the underlying processes are both driven by both theoretical and market developments. We clarify now this better in the introduction. 

“These market changes are strongly driven by theoretical developments, namely the development of large language models (Usman Hadi et al., 2023) and generative adversarial networks (Creswell et al., 2018). These new modeling developments have also led to a need to adjust traditional data driven master programs to encompass these new theoretical developments, and to integrate theoretical knowledge into interdisciplinary master programs. The rapid development of these new modeling approaches represents a continuous challenge to data science master programs to adopt their curricula.”

And also nuance the statement in the theory section:

“This growth has been driven by a combination of increasing market needs and theoretical development of the field of data science, as well as related fields such as business analytics or big data. Yet, the definition of the field is often driven by market needs and technology forecasts (for example Gorman & Kimberg, 2014). ”

2. Enhance the depth of your results section. It is crucial to delve deeper into the implications of findings, such as the omnipresence of the word “data” in all topics. Clarify the significance of such observations in differentiating and classifying courses. Additionally, it provides a clearer context, explanation, and discussion for the posterior distributions and their means, emphasizing their relevance in the distribution of topics across universities. 

We agree, and are thankful for this suggestion, indeed the word data is present in almost all clusters. That means that it is a very widely use word- and while overly relevant in all topics, less differentiating between clusters. We discuss the word clusters in more detail now in the result section: 

“When we look at the indicated clusters of skills in Table 3, we can clearly see that there is a wide variety of skills needed to complete a data science master program in the Netherlands. At the same time, there is also some overlap between the 7 topics. Most predominantly, the word “data” is present in every topic, not surprisingly, given its central role in these master programs. Also the words “model” (present in topics 3, 4, 5, 7), “Methods” (topics 3, 4, 7) and “machine learning” (topics 3, 4) are central to the clustering. Their presence in all these different clusters shows the wide spread importance of these skills. Each topic, however, has a specific focus characterized by the composition of the most frequent 10 words. For example, topic 2 is described by words like “statistical”, “theory”, “linear”, “data analysis”- that relate more to the more traditional statistical aspects of data science. In contrast, topic 3 focuses on “deep learning”, “natural language processing”, “machine learning”- more innovative elements of a broader data science field, while topic 5 encompasses skills like “health”, “ethical”, “decision”, pointing to applications of data science in different domains. We manually labelled each topic based on the 10 most frequent words and distinguished between core elements of data science and subdomains in Table 4 to aid interpretation.”

Furthermore with regard to the posterior distribution per universities we now updated figure 3, making it also colored, hopefully this improves the ease of readability. 

3. Expand the discussion to include a more detailed analysis of individual courses, considering they form the basic unit of your study. Investigate within-group variability, such as similarities and differences among courses and programs within the same university and how they compare to those between different universities. Incorporate hierarchical clustering analysis at multiple levels (courses, programs, universities) to provide a more comprehensive understanding.

 We thank the reviewer for this suggestion. We have added now a more detailed discussion of the 2 programs at Utrecht University and the two programs at Groningen university with the distribution of topics per program. We visualize this in Appendix E, and discuss it in the result section (see citation below). For this 4 programs we also add in Appendix F the per course distribution of the topics, while for all the courses this data is available in our R environment that we make available at OSF open publication of the materials. 

“Even within the same university, there can be large differences between skills taught, depending on the focus of the master programs. For illustrative purposes, Appendix E presents the posterior distribution of the seven topics across the two master programs offered by Utrecht and Groningen University. At Utrecht University, both the Applied Data Science and Artificial Intelligence programs have a similar focus on the research pillar. However, the Applied Data Science program has a heavier focus on core terms and statistics than the Artificial Intelligence program. At the same time, at Groningen University, the specialized Marketing Analytics and Data Science program shows less emphasis on the second to forth pillars than the general Artificial Intelligence program. Interestingly, the two AI master programs at these universities are more similar to each other than to the two more applied, specialized programs, and this pattern is also observed in the subject specific programs. 

To address Research Question 3, we examined the distribution of posterior classifications by program type, as depicted in Figure 3. This analysis distinguishes between general programs—with a broad focus on data science and AI skills—and subject-specific programs, which concentrate on a more narrow, domain-specific application of data science, such as in life sciences or marketing. Similar to the more detailed comparison in Appendix E, Figure 3 also shows a marked difference between general programs, which focus on the more technical pillars 3-5, and the subject-specific programs, which stand out in the first two, and last research pillars, thus the less technical pillars. “

 We are thankful for the suggestion of using a multilevel model, because at first instance we also considered the idea of modeling the multilevel structure in the data, however that was not possible for several reasons: namely i. lack of a large enough sample at university level (only 11 higher level units), and ii most importantly lack of proper multilevel models for clustering with unsupervised text mining using Latent variables for interpretable topics). 

We carefully checked other manuscripts using similar unsupervised text mining approaches to educational data (West, 2017 ) and also clustering for job market data (Verma et al, 2019, 2022, Almgerbi et al., 2022) and none of them applied a multilevel approach- because of a lack of such an approach. Also studies that applied a more qualitative approach ignore the multilevel structure, due to the complexity of properly analyzing it (Lee and Delaney, 2021, Gorman and Klimberg, 2014, Tang and Sae-Lim,2016). The reason for the lack of such proper multilevel models for these type of data probably lies in the sparsity of the word by document matrix that is used for the clustering. How to fit a multilevel structure on such sparse data is not obvious. We now added a paragraph in the Discussion where we clearly mention the lack of hierarchical approaches that are similar to multilevel regression in the field of multidimensional data reduction using unsupervised text mining approaches: 

“Another shortcoming of our approach is that although the data has a multilevel structure, with courses and programs nested within universities, we did not explicitly model this multilevel structure. The reason for this is that currently no topic modelling approach is available that can properly model the hierarchical structure in the data. ”

Furthermore we summarize basic descriptive statistics about the individual courses and programs in Table 1, where the number of programs and separately core and specialization courses per university are provided, and in appendix D we also provide the number of students per program. Appendix D gives a bit more insights about the variability with regard to the size of the programs as well, showing their importance on the market. 

4. Adjust the figures and their fonts to enhance readability. This is essential for you to convey your findings and facilitate reader comprehension.

We adjusted the figures. Figure 2 is now color and we changed the setup from having multiple barcharts stacked next to each other, to a single bar chart- so space is not wasted for showing the axis multiple times. Adding color to the graph also helps showing the differences between the universities. Furthermore we changed Figure 3 from a bar chart to a line-chart, because this occupies less space and highlights the differences even better. In appendix E we also added a new lien chart about the 2 programs at UU, and the 2 programs at RUG. 

5. Conduct a thorough spell check and grammar review. Address typographical errors like the one found in the first paragraph of section 3 (“correlated”). 

 Thank you for this comment. We revised the ms for grammar. 

6. Clarify inconsistencies in the number of Dutch universities included in the study. Could you make sure uniformity in the number of institutions referenced throughout the paper?

We would like to thank you for pointing out this inconsistency. We have 11 universities in the sample plus an MSc program that is organized in cooperation between 3 universities as a joint program. We now corrected this error in the abstract and on page 15. 

6. Refine your third research question to clearly distinguish between "general" and "subject-specific" data science programs. Providing specific examples or a more detailed explanation would enhance clarity.

Thank you for this comments. Indeed the differentiation was not clearly specified- we meant by general programs data science programs that are not field specific, and by subject-specific programs, programs that focus on data science within a disciplinary context for example Data science for marketing, or for the social sciences. – we now add an explanation on page 4 (see also citation below), and in the new appendix D in the last column we provide the classification for each master program. 

“(3) How do the skills learned at more general data science and AI programs differ from those learned at more subject-specific data science master programs? In this context, we define 'general data science and AI programs' (G) as programs exclusively focused on this domain. We refer to 'subject-specific programs' (S) as those that concentrate on domain-specific applications of data science, such as data science for marketing. ” 

Furthermore, we also discuss this in more detail in the results section, where the new Figure 3 zooms in more specifically on the distribution of the 7 topics over these two program types, see page 13: 

“To address Research Question 3, we examined the distribution of posterior classifications by program type, as depicted in Figure 3. This analysis distinguishes between general programs—with a broad focus on data science and AI skills—and subject-specific programs, which concentrate on a more narrow, domain-specific application of data science, such as in life sciences or marketing. Similar to the more detailed comparison in Appendix E, Figure 3 also shows a marked difference between general programs, which focus on the more technical pillars 3-5, and the subject-specific programs, which stand out in the first two, and last research pillars, thus the less technical pillars. 

Figure 3 The posterior distribution per program type (general vs subject specific) of the 7 topics of the CTM model. The classification of programs into general vs. subject specific are available in Appendix D.

Figure 3 illustrates a pronounced distinction between the two types of programs. General programs tend to score higher in the data processing clusters, particularly in research and electronics/IT. Conversely, subject-specific programs predominantly feature core terms related to statistics and ethics. A closer examination of the defining words within these clusters reveals that subject-specific programs not only emphasize data science technical skills but also integrate terms associated with ethics, health, and business. In contrast, general programs are characterized by a focus on technical and engineering skills.”

7. Standardize the terminology used throughout the paper. You can choose a consistent term (skills, competencies, knowledge) and give a clear definition to avoid ambiguity. 

We agree with the reviewer that the terms should be better standardized. We extracted the key terms directly from the data using unsupervised clustering some of the key terms are skills, while others fall under competencies or knowledge domains. We explicitly explain this issue in the current version on page 4: 

 “Moreover, it is important to clarify that our methodology employs unsupervised clustering combined with text mining techniques to derive a categorization of skills from the descriptions of the programs and courses. This approach ensures that the resulting clusters encapsulate a comprehensive range of knowledge, skills, and competencies (KSC), as characterized by Heller, Unlu, and Albert (2013). These clusters are not intended to segregate the three dimensions of KSCs; rather, we use the term 'skills' as an inclusive term that embodies all three dimensions. Our objective is to provide a holistic understanding of the KSCs imparted by these master programs, thereby refraining from distinguishing between knowledge, skills, and competencies. For the sake of brevity and clarity in the remainder of this article, we will refer collectively to these elements as 'skills', implicitly encompassing the full spectrum of knowledge, skills, and competencies as discussed. “

We use “skills” thereafter as the only general term. 

9. Elaborate on the inclusion criteria for the VSNU group of institutions. Could you clarify how these institutions compare to others and the representativeness of your sample?

We appreciate the reviewer's request for clarification regarding the inclusion of institutions from the VSNU group. The VSNU, now known as 'The Universities of the Netherlands', is a consortium recognized by the Dutch government that comprises all major government-funded universities in the Netherlands. This includes ten research universities, three specialized institutions, and one open university, which collectively represent 74% of the country's research universities. Membership in the VSNU assures adherence to stringent quality benchmarks and uniform academic standards across these institutions.

The excluded 26% of research universities consist of five small, specialized institutions: three theological universities dedicated to theological studies for clergy, one military university focused on defense education, and one private business school. Notably, these institutions do not offer master's programs in data science or artificial intelligence, which are the focus of our study.

To maintain the focus on the research scope and to avoid potential confusion, we have omitted the detailed information about the VSNU in our revised manuscript. We have instead emphasized that our analysis encompasses all Dutch master's programs in data science and AI for which course and/or program descriptions were publicly available, ensuring the reliability of our results. This refinement clarifies that the study is comprehensive within the context of the relevant educational programs in the Netherlands.

10. Add an annex listing the specific titles of data science programs per institution. Include the size of the corpus for each institution, which would enrich the context of your analysis. 

We added an extra table with this info into the new appendix D that shows all programs per university and the size of the corpus for the academic year 2020/21, the year of data collection. This table also has a column, that shows if a program was classified as general Data science/ AI (G) or domain specific (S) that helps interpreting the results for RQ3. 

11. Explore the inclusion of "soft skills" in Dutch data science Master's programs and discuss the implications of your findings in this regard.

Thank you for bringing this up, to motivate us to further elaborate. We actually found it concerning how using the text mining approach soft skills did not surface at all among the skills that define the clustering. We did a data driven approach- so this means this skills are not central enough/ not present to inform the clustering. We suggest for follow up research to look into more details about why these skills are not central. Follow up research should look into this very important issue in more detail, using different approaches- for example by interviewing program coordinators, because the course descriptions in their own did not provide sufficient data on this very important topic. We specifically discuss the problem in the discussion on page 16-17:

“A notable finding from our analysis is furthermore the absence of soft skills in the clustering results. Neither the 7-topic nor the more comprehensive 13-topic solutions revealed any reference to soft skills within the curricula. This suggests that soft skills are not a central or frequent component in the master programs we examined. While this finding aligns with previous research, which also highlights a predominant focus on technical skills (Tang and Sae-Lim, 2016; West, 2017; Cegielksi and Jones-Farmer, 2016), it remains a concerning trend, especially considering research that highlights the demand for soft skills in the job market (Allen et al., 2021; Mauro et al., 2017; Markow et al., 2017). In light of this discrepancy between curricular content and market needs, further investigation, possibly through qualitative methods such as interviews with program managers, is recommended to explore the reasons behind this lack of emphasis on soft skills.”

Your manuscript offers valuable insights into the field of data science education. I think addressing these recommendations will enhance your work's clarity, coherence, and overall impact. We are excited to receive your revised manuscript.

Sincerely,

***

---

## [Editor Report · Decision Letter 2]

9 Feb 2024

Unravelling the Skills of Data Scientists: A Text Mining Analysis of Dutch University Master Programs in Data Science and Artificial Intelligence

PONE-D-23-16749R2

Dear Dr. Bakk,

We’re pleased to inform you that your manuscript has been judged scientifically suitable for publication and will be formally accepted for publication once it meets all outstanding technical requirements.

Kind regards,

Ricardo Limongi

Academic Editor

PLOS ONE
---

## [Editor Report · Acceptance letter]

21 Feb 2024

PONE-D-23-16749R2 

PLOS ONE

Dear Dr. Bakk, 

I'm pleased to inform you that your manuscript has been deemed suitable for publication in PLOS ONE. Congratulations! Your manuscript is now being handed over to our production team.

Kind regards, 

on behalf of

Professor Ricardo Limongi 

Academic Editor

PLOS ONE